# Research on Hybrid Force Control of Redundant Manipulator with Reverse Task Priority

**DOI:** 10.3390/ma15196611

**Published:** 2022-09-23

**Authors:** Yu Su, Haiyan Liu, You Li, Bin Xue, Xianqing Liu, Minsi Li, Chunlan Lin, Xueying Wu

**Affiliations:** 1School of Mechanical and Transportation Engineering, Guangxi University of Science and Technology, Liuzhou 545006, China; 2Academic Affairs Office, Guangxi University of Science and Technology, Liuzhou 545006, China; 3School of Science, Guangxi University of Science and Technology, Liuzhou 545006, China; 4School of Automation, Guangxi University of Science and Technology, Liuzhou 545006, China

**Keywords:** task priority control, redundant robots, inverse kinematics, impedance control

## Abstract

This paper presents the reverse priority impedance control of manipulators with reference to redundant robots of a given task. The reverse priority kinematic control of redundant manipulators is first expressed in detail. The motion in the joint space is derived following the opposite order compared with the classical task priority–based solution. Then the Cartesian impedance control is combined with the reverse priority impedance control to solve the reverse hierarchical impedance controlled, so that the Cartesian impedance behavior can be divided into the primary priority impedance control and the secondary priority impedance control. Furthermore, the secondary impedance control task will not disturb the primary impedance control task. The motion in the joint space is affected following the opposite order and working in the corresponding projection operators. The primary impedance control tasks are implemented at the end, so as to avoid the possible deformations caused by the singularities occurring in the secondary impedance control tasks. Hence, the proposed reverse priority impedance control of manipulator can achieve the desired impedance control tasks with proper hierarchy. In this paper, the simulation experiments of the manipulator will verify the proposed reverse priority control algorithm.

## 1. Introduction

In spite of the past several decades of intense studies, inverse kinematics remains as one of the most fundamental issues of redundant manipulator research. Researchers have developed a great deal of methods, such as the position level inverse kinematics, velocity level kinematics and acceleration level kinematics [1,2,3]. Besides this, singularity robust control of redundant manipulators is also studied, such kinematics uncertainty and dynamics uncertainty. All of these studies aim to provide better control effects of manipulator [4].

The redundant manipulator has more flexibility and has more interest from scholars. The inverse kinematics of redundant manipulators aims to solve the relationship between the variables in the operational space and the variables in the joint space. With the complete utilization of the main DOFs (degrees of freedom), the residual DOFs are utilized to singularity, obstacle avoidance, multi-task optimization and so on [5,6,7]. Taking the joint limits avoidance index for example, the seminal work focused on the potential functions in order to complete the obstacle avoidance or limits avoidance efficiently [8]. Generally speaking, the multi-task optimization was always implemented in the null-space of the main tasks, so the frame of kinematics were based on the Jacobian of the manipulator.

Unlike the above solutions, another seminal work introduced the task priority–based solution. A primary task was implemented firstly, and a secondary task was implemented in the null space of the main task. Thus, the task accuracy of the primary task would be guaranteed, then the task accuracy of secondary task would be guaranteed as soon as possible [9,10]. Oussama Kanoun (2011) generalized the task priority framework of inequality tasks for kinematic control of redundant manipulators [11], and they also deal with the issue of inequality constraints by transformed the inequality constraints on equality constraints, and the potential fields are also introduced [12,13]. These approaches establish a strict hierarchy between tasks, i.e., tasks of lower priority do not affect the performance of the highest priority task, since they are projected in the null space of the kinematic Jacobian [14]. Chiaverini (1997) proposed the singularity-robust task-priority redundancy resolution, and it is used for real-time kinematic control of robot manipulators [15]. Gianluca (2009) also gave the stability analysis for task priority inverse kinematic solver [16]. A novel practical technique is proposed to integrate both the activation and deactivation of tasks and the inequality controlled objectives, and thus the discontinuity of enabling and disabling tasks [17,18]. Recently, the reverse priority approach was proposed to eliminate the algorithmic singularity and improve the task accuracy [19,20].

The above research results are focused on the position control of manipulators in the freedom space. As for the manipulator–environment interaction control, the impedance control was proposed [21,22]. Generally speaking, the impedance control can be divided as: typical impedance control, position–force hybrid control and hybrid impedance control [23,24,25]. Moreover, the impedance control can also be divided into the Cartesian impedance control and the joint impedance control, and in the Cartesian impedance control, the end-effector force/torque sensor was utilized, and in the joint impedance control algorithms, the joint torque sensor was utilized. The usage of the impedance relationship between the force/torque and the position response was to transform the dates from the sensors into the control input.

The force control of manipulators has been researched for many years. Hogan first proposed the concept of impedance control. It was also applied to the industry area. Heinrichs gave the position-based impedance control of an industrial hydraulic manipulator, and the nonlinear proportional-integral controller was developed to achieve the accurate positioning requirements [26]. Focchi (2016) proposed the inner torque and velocity feedback loops–based robot impedance control [27]. Lastly, the velocity feedback aimed to improve the bandwidth of the torque loop without the need of a complex controller.

The impedance control is a general approach to achieve tasks in which the robot behaves as a mass–spring–damper system whose inertia–damping–stiffness can be specified arbitrarily. The impedance control aimed to give a best solution to regulate the interaction force which may be changed due to the uncertainty of the location of the contact point and the environment’s structural properties, and the impedance controllers of manipulators have been widely utilized [28,29,30,31,32].

In this article, the concept of reverse priority forced control is proposed. The research topics on the proposed reverse priority force control aims to achieve the follow goals:(1)The implementation of hierarchical force control in the Cartesian space.(2)The algorithmic singularities can be reduced.(3)The manipulator–environment interaction can be guaranteed.

In this paper, we are willing to give a reverse priority impedance control law from the view of robotics. The paper is organized as follows: In Section 2, the inverse kinematics solutions are expressed. The reverse task priority inverse kinematics are described in Section 3. The reverse task priority impedance control of redundant manipulator is described in Section 4. Section 5 shows the simulation results. The conclusions is presented in Section 6.

## 2. Preliminaries

### 2.1. Jacobian-Based Solution

The inverse kinematics are based on the following Jacobian-based relationship [33,34]: (1)$$\dot{x}=Jq,$$
where $\dot{x}\in R^{m},\dot{q}\in R^{n}$, and $J\in R^{m\times n}$.

Considering about the least-square-based solution, the optimal issue can be listed as:(2)$$\min_{\dot{q}}∥\dot{x}-J\dot{q}∥,$$

The solution can be found by searching the best, that is:(3)$$\left(J^{T}J\right)\dot{q}=J^{T}\dot{x},$$

Thus, the pseudo-inverse solution of the above equation can be shown as:(4)$$\dot{q}={\left(J^{T}J\right)}^{-1}J^{T}\dot{x},$$

The above equation stands for the end-effector position and posture control. It is necessary to add the residual arbitrariness in the above equation to give a general expression including the null space. Hence, the total manifold can be shown as:(5)$$\dot{q}=J^{+}\dot{x}+\left(I-J^{+}J\right)\dot{\zeta },$$
where $\dot{\zeta }$ is the arbitrary null space vector. The additional multi-task optimization can be implemented in the null vector by the above equation.

However, the above equation ignored the ill condition of Jacobian matrix. Furthermore, the regularization equation can be modified by adding additional regularization cost such as:(6)$$\min_{q}∥\dot{x}-J\dot{q}{∥}^{2}+{∥\dot{q}∥}_{\dot{\lambda }}^{2},$$
where λ≥0 is the weighted matrix, and $∥\dot{q}{∥}_{\lambda }^{2}$ is the weighted norm, and $∥\dot{q}{∥}_{\lambda }^{2}={\dot{q}}^{T}\lambda \dot{q}$.

The solution of the above equation can be shown as:(7)$$\dot{q}={\left(J^{T}J+\lambda \right)}^{-1}J^{T}\dot{x}+\left(I-{\left(J^{T}J+\lambda \right)}^{-1}\left(J^{T}J+\lambda \right)\right)\dot{\zeta },$$

As for the redundant manipulator, the null space vector stands for the gradient direction of position-dependent scalar index such as the joint limits avoidance:(8)$$H\left(\theta \right)=\sum \left(\frac{\theta_{i}-\theta_{i_{-}\mathrm{mid}}}{\theta_{i_{-}\mathrm{mid}}-\theta_{i_{-}\max}}\right).$$

### 2.2. The Task-Priority-Based Solution

In the above Jacobian-based solution of redundant manipulator, the optimization task are implemented in the null-space of the main task. The task hierarchy inverse kinematics is well-established on the following task hierarchy forward kinematics as:(9)$${\dot{x}}_{i}=J_{i}\dot{q},\;i=1,2,$$
where and represent task1 and task2, respectively.

The inverse kinematics of redundant manipulator that learned from expression (5) can be expressed as:(10)$$\dot{q}=J_{i}^{+}{\dot{x}}_{i}+\left(I-J_{i}^{+}J_{i}\right)\dot{\zeta },$$

Implementing task1 as the primary task, and task2 as the secondary task, the secondary impedance control task will not disturb the primary impedance control task [35,36]. That is to say, task2 is implemented in the null-space of task1. The final inverse kinematics expression of the redundant manipulator is as follows:(11)$$\dot{q}=J_{1}^{+}{\dot{x}}_{1c}+N_{1}{\bar{J}}_{2}^{+}\left({\dot{x}}_{2c}-J_{2}J_{1}^{+}{\dot{x}}_{1c}\right)+N_{2}\dot{\zeta },$$
where $N_{1}=\left(I-J_{1}^{+}J_{1}\right)$ and $N_{2}=N_{1}\left(I-{\bar{J}}_{2}^{+}J_{2}\right)\cdot {\bar{J}}_{2}=J_{2}N_{1}$ are the projected matrix, which gives the available range of the secondary task to the primary task. ${\dot{x}}_{1c}$ and ${\dot{x}}_{2c}$ are the desired command velocities. Generally speaking, ${\dot{x}}_{1c}$ is the primary task, and ${\dot{x}}_{2c}$ is the secondary task.

If the two related generic tasks are dependent, then the corresponding Jacobian matrix will be singular. If the task Jacobian is singular, the corresponding task will not be satisfied. In this case, the Jacobian-related matrix will be singularity, and this is defined as algorithmic singularity. Namely, the two generic tasks are dependent if
(12)$$\rho \left(J_{1}\right)+\rho \left(J_{2}\right)>\rho \left(\left[\begin{array}{c} J_{1}\\ J_{2} \end{array}\right]\right),$$
where ρ([ ]) is the rank of a matrix.

It is obvious that the algorithmic singularity is introduced by the task conflict between the secondary and the primary task. Moreover, the task-priority-based inverse kinematics of redundant manipulator aims to give a better control effect of the primary task; for example, let the position control direction as the primary task, and then the position control direction task accuracy will be guaranteed. In a prior research’s results, Chiaverini proposed the singularity robust solution to give an algorithmic singularity-free algorithm, and the two priority tasks are given as follows:(13)$$\dot{q}=J_{1}^{+}{\dot{x}}_{1}+\left(I-J_{1}^{+}J_{1}\right)J_{2}^{+}{\dot{x}}_{2}.$$

The algorithmic singularity is eliminated by the special mathematical derivation.

### 2.3. Singularity-Robust Solution

In the classical Jacobian pseudo-inverse–based solution will occur the kinematics singularity, and this is caused by the matrix under-rank. It is necessary to give the damped least squares (DLS) solution for the issue of kinematic singularity.

As for the DLS solution, the general cost function can be modified as:(14)$$\min_{q}∥\dot{x}-J\dot{q}{∥}^{2}+\eta^{2}{∥\dot{q}∥}^{2},$$

Hence, the singularity robust pseudo-inverse solution of the above equation can be shown as:(15)$$\dot{q}={\left(J^{T}J+\eta^{2}I\right)}^{-1}J^{T}\dot{x},$$

Let $\lambda =\eta^{2}I$ , and the above DLS solution is equivalent to the additional regularization solution. The scalar value balances the task accuracy and the singularity.

As for the calculation of the Jacobian matrix pseudo-inverse, we can give the singular value decomposition (SVD) form of the Jacobian:(16)$$J=U\Sigma V^{T},$$
where $U\in R^{m\times m},\;V\in R^{n\times n}$ , $\Sigma \in R^{m\times n}$ and U is a unitary matrix composed by column vectors $u_{i}\cdot V$ is a unitary matrix composed by column vectors $v_{i}\cdot \sum$ is a block matrix with a leading m×m diagonal matrix containing the singular values $\sigma_{i}\geq 0\;\mathrm{of}\;J$ in decreasing order following by n−m zero columns.
(17)$$J^{+}=V\Sigma^{-1}U^{T}=\sum_{i=1}^{r}\frac{1}{\sigma_{i}}v_{i}u_{i}^{T},$$
where r≤m is the rank of *J* .

As for the kinematic singularity, referring to the SVD used for computing the needed pseudo-inverse, the large generated joint velocities are due to the smallest singular value rapidly approaching 0. The approach is followed as:(18)$$\frac{1}{\sigma_{i}}=\frac{\sigma_{i}}{\sigma_{i}^{2}+\lambda_{0}^{2}},$$

The factor $\lambda_{0}$ will affect the singularity. The higher the $\lambda_{0}$ is, the more damped are the joint velocity near singularity. Besides this, there are different methods of defining a variable damping factor. Then we have:(19)$$\lambda_{0}=\left\{\begin{array}{c} 0\;\;\;\;\;\sigma_{m}\geq \delta \;\\ \left(1-{\left(\frac{\sigma_{m}}{\delta }\right)}^{2}\right)\lambda_{\max}^{2}\;\;\mathrm{otherwise}\;. \end{array}\right.$$

From the above equation, we can see that the parameter δ>0 monitors the smallest singular value.

## 3. Reverse Priority Control of Manipulator

The classical Jacobian pseudo-inverse–based solution can not give a task hierarchical control. That is to say, all tasks are in the same priority, and then DLS-based singularity-robust solution will sacrifice the accuracy of all tasks. Moreover, the classical task-priority-based solution will cause algorithmic singularity. So it is necessary to introduce a novel inverse kinematic solver for handling multi-robotic-task with hierarchy that is unaffected by algorithmic singularity, and the task accuracy will also be guaranteed in case of kinematic singularity. Thus, in the following part, we give the reverse priority approach to solve the kinematic issue of redundant manipulator.

### 3.1. Reverse Priority Control of Redundant Manipulator with Multiple Tasks

To elaborate the reverse priority kinematic control of redundant manipulator, it is necessary to introduce the concept of reverse priority projection matrix ${\overset{˘}{P}}_{k+1}$. This matrix includes the l−k−1 tasks of the lowest priority that are not dependent from the *k*-th tasks. Hence, we have:(20)$${\overset{˘}{P}}_{k+1}=I-{\overset{˘}{J}}_{k+1}J_{k+1},$$
(21)$${\overset{˘}{J}}_{k+1}={\left(J_{k+1|k}^{T}J_{k+2|k}^{T}\cdots J_{1|k}^{T}\right)}^{T},$$
where $J_{i|j}$ is the Jacobian associated to all components of the *i*-th tasks that are linearly independent from the *j*-th task.

The reverse priority recursive formula is as follows:(22)$${\dot{q}}_{k}={\dot{q}}_{k+1}+{\left(J_{k}{\overset{˘}{P}}_{k+1}\right)}^{+}\left({\dot{x}}_{k}-J_{k}{\dot{q}}_{k+1}\right),$$

In the above derivation, k=l,l−1,⋯,1. The initialization value is ${\dot{q}}_{l+1}=0$.

To give a general calculation form of the linearly independent Jacobians $J_{i|j}$ , we define the reverse augmented Jacobian matrix as:(23)$$J_{k}^{RP}={\left(J_{k}^{T}J_{k+1}^{T}\cdots J_{1}^{T}\right)}^{T}=\left(\begin{array}{c} J_{k}\;\\ J_{k+1}^{RP}, \end{array}\right)$$

Then we have:(24)$$\prod_{k+1}J_{k=1}^{RP}=\left(\begin{array}{c} {\overset{˘}{J}}_{k+1}\;\\ {\bar{J}}_{k+1}, \end{array}\right)$$
where ${\bar{J}}_{k+1}$ denotes the row of $J_{k=1}^{RP}$ .

Furthermore the pseudo-inverse of can be partitioned as:(25)$$J_{k}^{RP_{+}}=\left(T_{k},H_{k}\right),$$
(26)$$H_{k}=J_{k}^{RP+}-T_{k}J_{k}J_{k+1}^{RP+},$$
where $T_{k}$ denotes the augmentation of $J_{k+1}^{RP+}$ .

The final reverse priority projector can be written as:(27)$${\overset{˘}{P}}_{k+1}=I-\left({\overset{˘}{J}}_{k+1}^{+}0\right)\left(\begin{array}{c} {\overset{˘}{J}}_{k+1}\;\\ {\bar{J}}_{k+1} \end{array}\right)=I-\left(I-T_{k}T_{k}^{+}\right)H_{k}J_{k+1}^{RP},$$

Then we have the expression of the pseudo-inverse calculation:(28)$$\begin{array}{c} {\left(J_{k}{\overset{˘}{P}}_{k+1}\right)}^{+}\&={\left(J_{k}\left(P_{k}^{RP}+T_{k}T_{k}^{+}\right)\right)}^{+}\\ ={\left(J_{k}P_{k}^{RP}+J_{k}T_{k}T_{k}^{+}\right)}^{+}\\ ={\left(J_{k}T_{k}T_{k}^{+}\right)}^{+}\\ =T_{k}{\left(J_{k}T_{k}\right)}^{+} \end{array}$$

In summary, the reverse priority calculation can be followed as:(29)$${\dot{q}}_{k}^{RP}={\dot{q}}_{k+1}^{RP}+T_{k}{\left(J_{k}T_{k}\right)}^{+}\left(\dot{x}-J_{k}{\dot{q}}_{k+1}^{RP}\right).$$

### 3.2. Reverse Priority Control of Redundant Manipulator with Two Tasks

As for the 6-DOF manipulator or 7-DOF redundant manipulator, there are no abundant DOFs to achieve many hierarchical tasks. It is necessary to give the two-task priority control. That is to say, the classical kinematic control of the manipulator is just the primary task and the secondary task.

From the basic mathematical derivation, we can see:(30)$$\dot{q}=J_{2}^{+}{\dot{x}}_{2}+{\left(J_{1}{\overset{˘}{P}}_{2}\right)}^{+}\left({\dot{x}}_{1}-J_{1}J_{2}^{+}{\dot{x}}_{2}\right),$$

The above formula is significantly different from the previous expression (Equation 11), but the algorithm framework is similar. In the above equation, ${\dot{x}}_{2}$ is the secondary task, and ${\dot{x}}_{1}$ is the primary task. It is also obvious that the secondary task is calculated firstly. The primary task ${\dot{x}}_{2}$ is implemented in the specified null-space of the primary task. The kernel point of the reverse priority is the calculation of the projection matrix ${\overset{˘}{P}}_{2}$. The expression of ${\overset{˘}{P}}_{2}$ can be as follows:(31)$${\overset{˘}{P}}_{2}=I-{\overset{˘}{J}}_{2}J_{2},$$

Drawing on previous similar derivations from (Equation 23)–(Equation 29), we can also obtain the concise expression:(32)$$\dot{q}=J_{2}^{+}{\dot{x}}_{2}+T_{1}{\left(J_{1}T_{1}\right)}^{+}\left({\dot{x}}_{1}-J_{1}J_{2}^{+}{\dot{x}}_{2}\right).$$

## 4. Reverse Priority Impedance Control

The Cartesian impedance control can be divided into three cases. The first one is the Cartesian force tracking control, and this case can be applied to the human–manipulator interaction behavior, like the handshake, leg follower and so on. The secondary one is the Cartesian impedance control, and this case can be applied to the target operation control. The manipulator can track a desired path, and prevent itself from being hit by external objects. The last case is the Cartesian hybrid impedance control, and this case can be applied to a special operation or capturing situations.

### 4.1. The Reverse Priority Force Control of Manipulator

The dynamics of manipulator in the force control space can be written as follows:(33)$$M\left(X\right)\ddot{X}+H\left(X,\dot{X}\right)=F-F_{e},$$
where *X* is the position in the Cartesian space, M(X) is the inertia matrix, $H(X,\dot{X})$ is the nonlinear force, *F* is the input control force, and is the contact force.

Furthermore, the input joint torque can be obtained by the Jacobian matrix–based transformation:(34)$$\tau =J^{T}\left(q\right)F,$$

The desired motion equation of the manipulator in the force control space can be defined as follows:(35)$$M_{d}\ddot{X}+B_{d}\dot{X}-F_{d}=-F_{e},$$
where $M_{d}$ and $B_{d}$ are the inertia and damping matrix. $F_{d}$ is the commanded force and $F_{e}$ is the contact force. The scheme map of force control is shown in Figure 1:

The relationship between the environment and the manipulator response can be written as:(36)$$M_{e}\ddot{X}+B_{e}\dot{X}+K_{e}X=F_{e},$$

The combination of the above two equations can be followed as:(37)$$\left(M_{d}+M_{e}\right)\ddot{X}+\left(B_{d}+B_{e}\right)\dot{X}+K_{e}X=F_{d}.$$

From the above equation, we can see that if the $M_{e}$ ,$B_{e}$ and $K_{e}$ are known, the adjustment of $M_{d}$ and $B_{d}$ will influence the system response.

The force control makes it so that the manipulator can interact with the environments or human beings. Moreover, in some situations, it is not necessary to implement the force control in all directions, or it is not necessary to guarantee the force control in all directions, that is to say, sometimes, we only want to guarantee the force tracking control accuracy in some directions.

For example, if the manipulator interacted with a planer, it is only necessary to keep the accurate force tracking control in the vertical direction, and the other direction does not need accurate force tracking control. In other cases, the position–direction force control is more important than the pose direction force control. Thus, it is necessary to provide a hierarchy force control of the manipulator. That is to say, the novel framework of the hierarchy force control is necessary.

From the above equations we can obtain the desired hierarchy force control relationship as follows:(38)$$\begin{array}{c} M_{d1}{\ddot{X}}_{1}+B_{d1}{\dot{X}}_{1}-F_{d1}=-F_{e1}, \end{array}$$
(39)$$\begin{array}{c} M_{d2}{\ddot{X}}_{2}+B_{d2}{\dot{X}}_{2}-F_{d2}=-F_{e2}. \end{array}$$

The integral formula of these two equations can be written as:(40)$$\begin{array}{c} {\dot{X}}_{1c}=\int M_{d1}^{-1}\left(-F_{e1}+F_{d1}-B_{d1}{\dot{X}}_{1}\right)dt, \end{array}$$
(41)$$\begin{array}{c} {\dot{X}}_{2c}=\int M_{d2}^{-1}\left(-F_{e2}+F_{d2}-B_{d2}{\dot{X}}_{2}\right)dt. \end{array}$$

If the manipulator end-effector can track the desired Cartesian velocity as ${\dot{X}}_{1c}$ and ${\dot{X}}_{2c}$, then the accurate force control of manipulator can be implemented. The relationship between the Cartesian velocity and the joint velocity should learn from the reverse priority control. Hence, the reverse priority control of manipulator can be obtained as:(42)$$\begin{array}{c} \dot{q}=J_{2}^{+}\int M_{d2}^{-1}\left(-F_{e2}+F_{d2}-B_{d2}{\dot{X}}_{2}\right)dt\\ +{\left(J_{1}{\overset{˘}{P}}_{2}\right)}^{+}\left(\begin{array}{c} \int M_{d1}^{-1}\left(-F_{e1}+F_{d1}-B_{d1}{\dot{X}}_{1}\right)dt\\ -J_{1}J_{2}^{+}\int M_{d2}^{-1}\left(-F_{e2}+F_{d2}-B_{d2}{\dot{X}}_{2}\right)dt. \end{array}\right) \end{array}$$

The desired joint velocity as in the above equation will guarantee the force control of the manipulator. It is worth mentioning that the above force control law is just velocity level control law, and it is dependent on the inner-velocity-loop control. If the inner position control has a good effect, then the accurate force control will be implemented. As the inner-velocity-loop control can achieve a low frequency position tracking, the outer-force loop can also achieve a low frequency force tracking.

### 4.2. The Reverse Priority Impedance Control of Manipulator

When the manipulator implements the force control, the manipulator acts as the initiator to some degree, that is to say, the manipulator has already been prepared for responding to the external environment. If the manipulator is working as the impedance control model, the manipulator passively responds to external forces. The following Figure 2 shows the dynamics of this behavior.

The corresponding impedance relationship between the external force and the joint acceleration can be shown as
(43)$$\begin{array}{c} M_{d1}\left({\ddot{X}}_{1}-{\ddot{X}}_{1}^{d}\right)+B_{d1}\left({\dot{X}}_{1}-{\dot{X}}_{1}^{d}\right)+K_{d1}\left(X_{1}-X_{1}^{d}\right)=-F_{e1}, \end{array}$$
(44)$$\begin{array}{c} M_{d2}\left({\ddot{X}}_{2}-{\ddot{X}}_{2}^{d}\right)+B_{d2}\left({\dot{X}}_{2}-{\dot{X}}_{2}^{d}\right)+K_{d2}\left(X_{2}-X_{2}^{d}\right)=-F_{e2}, \end{array}$$

The reference velocity can be shown as:(45)$${\dot{X}}_{1ref}=\int \left(M_{d1}^{-1}\left(\begin{array}{c} -B_{d1}\left({\dot{X}}_{1}-{\dot{X}}_{1}^{d}\right)-\\ K_{d1}\left(X_{1}-X_{1}^{d}\right)-F_{e1} \end{array}\right)+{\ddot{X}}_{1}^{d}\right)dt,$$
(46)$${\dot{X}}_{2ref}=\int \left(M_{d2}^{-1}\left(\begin{array}{c} -B_{d2}\left({\dot{X}}_{2}-{\dot{X}}_{2}^{d}\right)-\\ K_{d2}\left(X_{2}-X_{2}^{d}\right)-F_{e2} \end{array}\right)+{\ddot{X}}_{2}^{d}\right)dt,$$

The reverse priority impedance control of manipulator can be obtained as:(47)$$\begin{array}{c} \dot{q}=J_{2}^{+}\int \left(M_{d2}^{-1}\left(\begin{array}{c} -B_{d2}\left({\dot{X}}_{2}-{\dot{X}}_{2}^{d}\right)-K_{d2}\left(X_{2}-X_{2}^{d}\right)-F_{e2} \end{array}\right)+{\ddot{X}}_{2}^{d}\right)dt\\ +{\left(J_{1}{\overset{˘}{P}}_{2}\right)}^{+}\left(\begin{array}{c} \int \left(M_{d1}^{-1}\left(\begin{array}{c} -B_{d1}\left({\dot{X}}_{1}-{\dot{X}}_{1}^{d}\right)-K_{d1}\left(X_{1}-X_{1}^{d}\right)-F_{e1} \end{array}\right)+{\ddot{X}}_{1}^{d}\right)dt\\ -\int \left(M_{d2}^{-1}\left(\begin{array}{c} -B_{d2}\left({\dot{X}}_{2}-{\dot{X}}_{2}^{d}\right)-K_{d2}\left(X_{2}-X_{2}^{d}\right)-F_{e2} \end{array}\right)+{\ddot{X}}_{2}^{d}\right)dt \end{array}\right). \end{array}$$

The above desired joint velocity can guarantee the desired impedance behavior of the manipulator in this case. Moreover, the velocity-based impedance control law will be easily adopted.

### 4.3. The Reverse Priority Hybrid Impedance Control of Manipulator

The hybrid impedance application is the combination of the above two, that is to say, the Cartesian task can be divided into two cases: the first is the position control subspace, and the impedance control is implemented in this subspace; the second is the force control subspace, and the force control is implemented in this subspace. Hence, a selection matrix is selected. The relationship between the external force and the position response is as follows:(48)$$\begin{array}{c} M_{1d}\Delta {\ddot{X}}_{1}+B_{1d}\Delta {\dot{X}}_{1}+K_{1d}S_{1}\Delta X_{1}=\left(I-S_{1}\right)F_{1d}-F_{1e}, \end{array}$$
(49)$$\begin{array}{c} M_{2d}\Delta {\ddot{X}}_{2}+B_{2d}\Delta {\dot{X}}_{2}+K_{2d}S_{2}\Delta X_{2}=\left(I-S_{2}\right)F_{2d}-F_{2e}. \end{array}$$

So the simplified form of the desired velocity can be shown as:(50)$$\begin{array}{c} {\dot{X}}_{1ref}=\int \left(\Delta {\ddot{X}}_{1}+{\ddot{X}}_{1}^{d}\right)dt, \end{array}$$
(51)$$\begin{array}{c} {\dot{X}}_{2ref}=\int \left(\Delta {\ddot{X}}_{2}+{\ddot{X}}_{2}^{d}\right)dt. \end{array}$$

Then we have the reverse priority-based solution:(52)$$\begin{array}{cc} \dot{q} & =J_{2}^{+}{\dot{x}}_{2ref}+{\left(J_{1}{\overset{˘}{P}}_{2}\right)}^{+}\left({\dot{x}}_{1ref}-J_{1}J_{2}^{+}{\dot{x}}_{2ref}\right)\\ \; & =J_{2}^{+}{\dot{x}}_{2ref}+T_{1}{\left(J_{1}T_{1}\right)}^{+}{\left({\dot{x}}_{1ref}-J_{1}J_{2}^{+}{\dot{x}}_{2ref}\right)}^{\prime }. \end{array}$$

The scheme map is shown in Figure 3:

Regarding n-hierarchy tasks, the corresponding impedance control tasks also belong to the n-hierarchy framework, so the general framework of the velocity-level reverse priority impedance control of the manipulator can be written as follows:(53)$$\begin{array}{c} {\dot{q}}_{k}=J_{k+1}^{+}{\dot{x}}_{k+1_{-}ref}+{\left(J_{k}{\overset{˘}{P}}_{k+1}\right)}^{+}\left({\dot{x}}_{k+1_{-}ref}-J_{k}J_{k+1}^{+}{\dot{x}}_{k+1_{-}ref}\right)\\ =J_{k+1}^{+}{\dot{x}}_{k+1}{}_{-1}{}^{k+}+T_{k}{\left(J_{k}T_{k}\right)}^{+}\left({\dot{x}}_{k+1}\;\mathrm{ref}\;-J_{k}J_{k+1}^{+}{\dot{x}}_{k+1_{-}ref}\right). \end{array}$$

The main work of the mathematical derivation of the reverse priority impedance control law in the above expressions aims to expand the reverse priority calculation of the position control space into that of the force control space. Moreover, the reverse priority impedance control framework derived in this paper will also be applied to the underwater manipulator, space manipulator and mobile robot. In addition, the velocity-level impedance control law will also be suitable for the real controller of the manipulator. The reverse priority impedance control gives an algorithmic singularity free impedance control, and the primary impedance control task will be guaranteed. The strict reverse priority will also give a better control effect of the manipulator.

## 5. Simulation

To verify the proposed control laws, it is necessary to build the simulation system to give a task verification. The corresponding manipulator system is 7-DOF sawyer manipulator. The total mass of this manipulator is 11 kg, and its payload is 4 kg. Table 1 gives the DH parameters, and the manipulator sketch map is shown in Figure 4.

### 5.1. Example 1

In this example, the comparison between the reverse priority forced control and the Jacobian-based force control is made. The effect of the force tracking down the proposed reverse priority force control is also given. Assume the desired force was chosen as follows:(54)$$F_{d}=30+5\sin\left(t/\mathrm{pi}\right)\left(N\right).$$

In the proposed reverse priority force control of manipulator, assume the position control direction X is primary force control task, and the Y and Z directions and all the pose control directions are the secondary force control tasks. The desired impedance control parameters were set as $M_{d}=\mathrm{diag}\left(15\;15\;15\;80\;80\;80\right)$, and the damping are $B_{d}=\mathrm{diag}\left(150\;150\;150\;280\;280\;280\right)$. For the convenience of explanation, we take X and Y directions as an example. Figure 5 shows the RP force control. And the classical force control is showed in Figure 6.

From the above Figure 5 and Figure 6, we can see that the manipulator can track the desired alternating force. It is worth mentioning that the alternating force supposed in this paper is a low frequency signal. This takes into account the bandwidth of the PID controller. Besides this, the Fx in the Figure 5 is obviously better than that in the Figure 6. This is because the Fx is the primary forced control task in the reverse priority force control algorithm.

### 5.2. Example 2

In this example, we will give the experiments about approaching and contacting a stiff wall. The control period and control parameters are as in the first experiment. Moreover, the aluminum alloy wall was used as the environment object. The stiffness is about $10^{5}\;N/m$, and the wall is assumed in the X direction. The whole process is an approaching-contact-stability process. Approaching object is an important behavior of automatic tasks with manipulators. The corresponding control results are shown in Figure 7 and Figure 8 (case 1 is the classical impedance control, case 2 is the reverse priority impedance control, and the X direction is the primary force control direction).

It can be seen from the above curve that the robot arm can achieve close approaching operation well via the proposed reverse priority impedance control law. The curves in case 2 overwhelm the curves in case 3. This demonstrates that the reverse priority impedance control indeed overwhelmed the classical impedance control algorithm. The final contact force in X direction is 30 N; the manipulator contacted the wall with a stable force.

## 6. Conclusions

This article has introduced the reverse priority force control algorithm to handle multi-force-control-task with strict priority. The novel method considers the primary priority force control task after the calculation of the secondary force control task. Moreover, the reverse control framework guaranteed that there are no algorithmic singularities. The task hierarchy in the force controller is also implemented. The primary force control task will first be guaranteed, and the secondary force control task can be implemented as soon as possible. Furthermore, the task hierarchy will also improve the task accuracy in the Cartesian force controller. The strict mathematical derivation of the reverse priority force control is obtained. The proposed reverse priority force control of the manipulator gives the algorithmic singularity free force control, and the completeness and validity of the theory are validated in the simulation.

## Figures and Tables

**Figure 1 materials-15-06611-f001:**
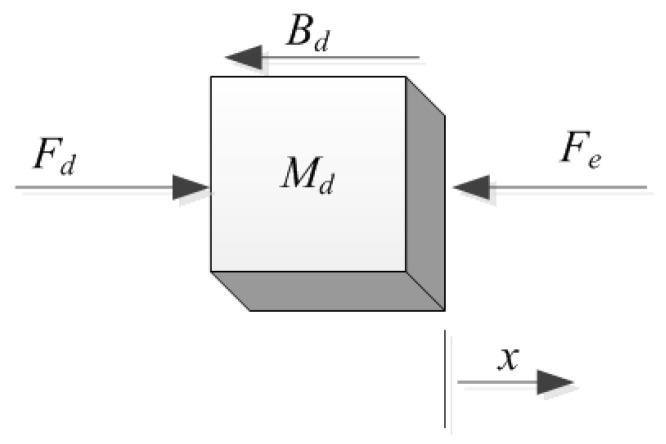
The dynamics of the force control.

**Figure 2 materials-15-06611-f002:**
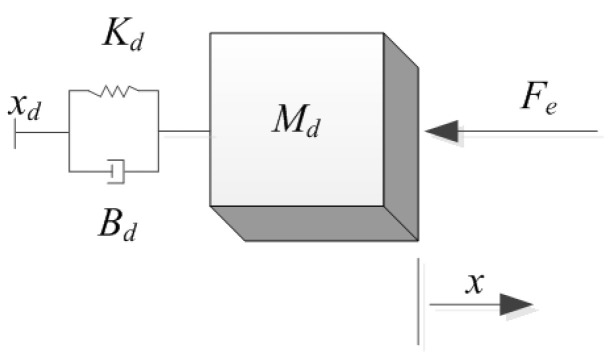
The dynamics of the impedance control.

**Figure 3 materials-15-06611-f003:**
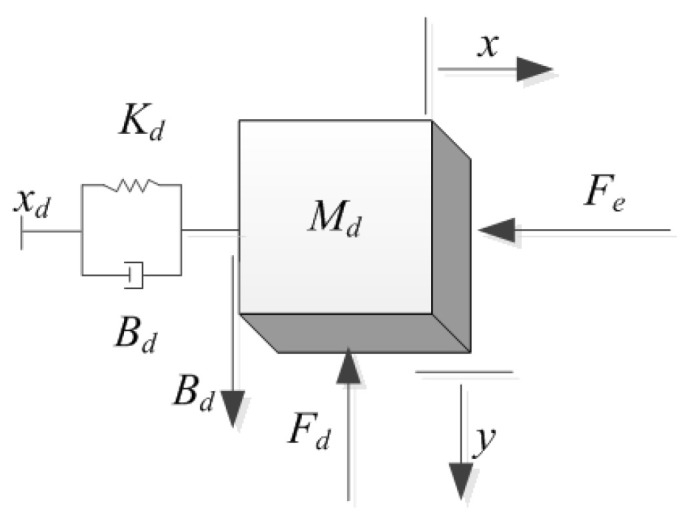
The dynamics of the hybrid impedance control.

**Figure 4 materials-15-06611-f004:**
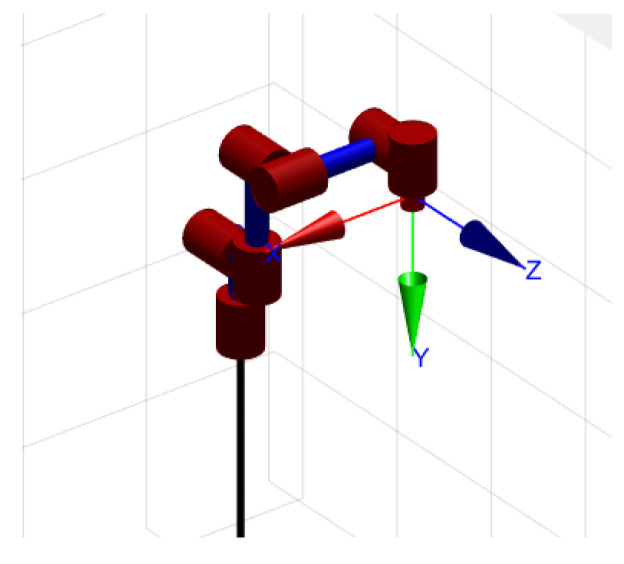
The manipulator sketch map.

**Figure 5 materials-15-06611-f005:**
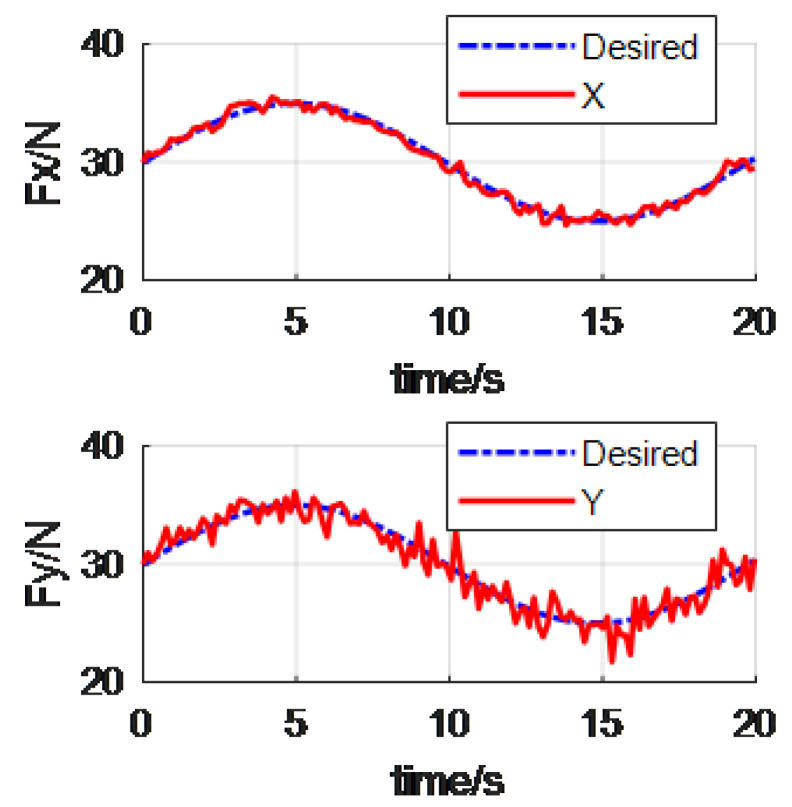
The tracking force via the RP force control.

**Figure 6 materials-15-06611-f006:**
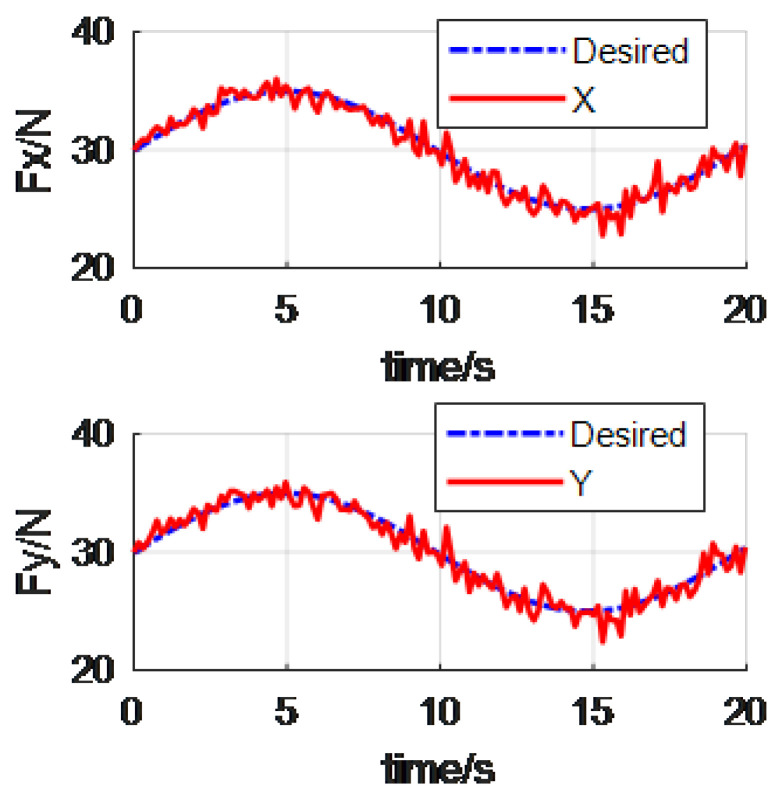
The tracking force via the classical force control.

**Figure 7 materials-15-06611-f007:**
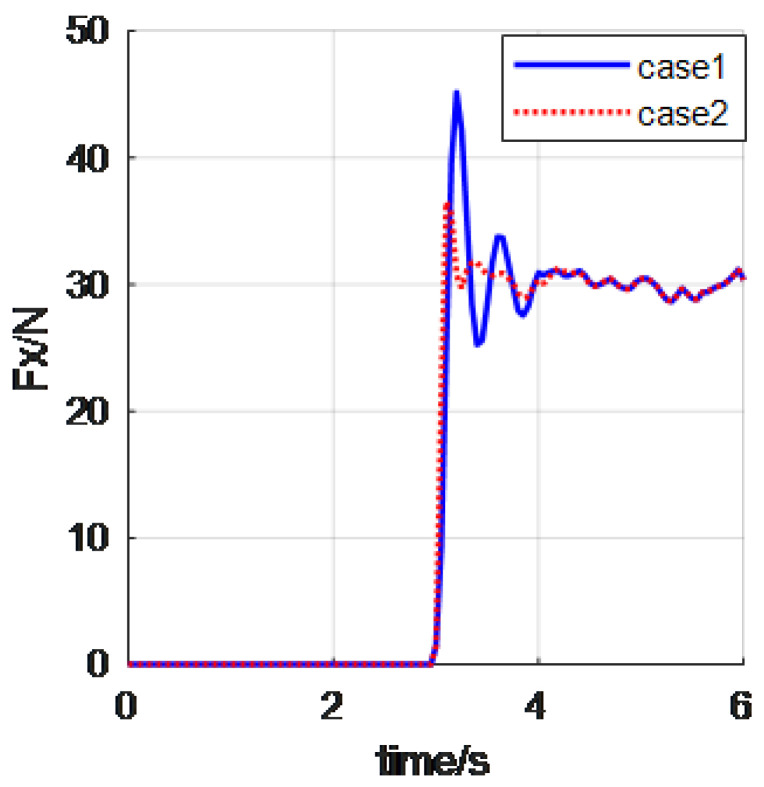
The contcact force.

**Figure 8 materials-15-06611-f008:**
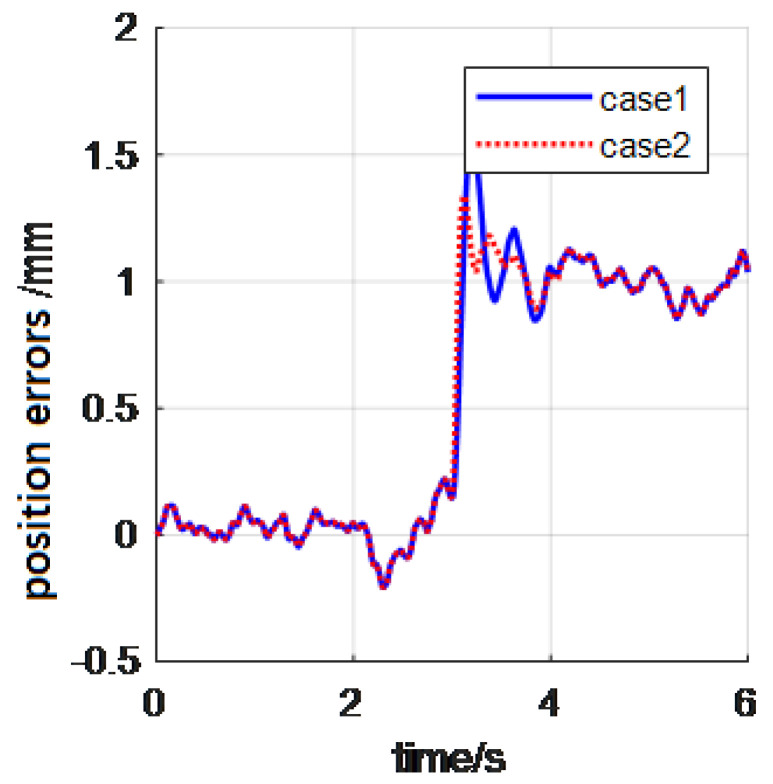
The position tracking errors.

**Table 1 materials-15-06611-t001:** The DH table of evaluation of sawyer.

	d	a	alpha	qlim
Link1	0.317	−0.081	−pi/2	[−pi,pi]
Link2	−0.1925	0	pi/2	[−pi,pi]
Link3	0.4	0	−pi/2	[−pi,pi]
Link4	−0.1685	0	pi/2	[−pi,pi]
Link5	0.4	0	−pi/2	[−pi,pi]
Link6	−0.1363	0	pi/2	[−pi,pi]
Link7	0.13375	0	pi/2	[−pi,pi]

## Data Availability

Not applicable.

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
