# Peer review of "Research on Hybrid Force Control of Redundant Manipulator with Reverse Task Priority"

_materials, 2022, doi:10.3390/ma15196611_

Round 1

Reviewer 1 Report

The figures did not appear in the article, which made it difficult to understand the manuscript. Please review all figures.

There are some typographical errors, please review the manuscript. References could include more current references such as 2020, 2021 and 2022.

Overall, the article is interesting but a comparative discussion of the results should be done for a deeper discussion of the manuscript.

Author Response

Dear Reviewer:

Thank you for your decision and constructive comments on our manuscript. We have carefully considered the suggestion of Reviewer and make some changes. We have tried our best to improve and made some changes in the manuscript.

The red part that has been revised according to your comments. Revision notes, point-to-point, are given as follows:

1. The figures did not appear in the article, which made it difficult to understand the manuscript. Please review all figures.

Thank you for the comment suggested.

We reviewed  and checked all figures. Please refer to the attachment.

2. There are some typographical errors, please review the manuscript.

Thank you for the comment suggested.

We reviewed  our manuscript and modified some typographical errors with red words. Please refer to the attachment.

3. References could include more current references such as 2020, 2021 and 2022.

Thank you for the comment suggested.

We  added more current references with red words. Please refer to the attachment.

Reviewer 2 Report

The paper titled, ‘Research on Hybrid Force Control of Redundant Manipulator with Reverse Task Priority’ proposes  the reverse priority impedance control of manipulators with reference to redundant robots of a given task, which has been verified using simulation work. The reverse priority impedance is interesting field and currently in great focus by the research community. However, following suggestions/comments will further improve the work, if incorporated.

1.     the secondary impedance control  task will not disturb the primary impedance control task; authors should provide reason(s) along with relevant literature work.

2.     Equation (1) is very fundamental concept and should be added with a citation.

3.     Authors are highly recommended, they add the table of comparison where they can compare their proposed algorithm with the recently reported works.

4.     There are many language/grammatical errors, which should be corrected.

5.     In whole manuscript, all the figures are just white screen; they are not visible; it may be at the end of journal portal or submission itself; so I am unable to see any results provided.

6.     Recent most references are not provided; authors are encouraged to add recent citations to support the study. 

Author Response

Dear Reviewer:

Thank you for your decision and constructive comments on my manuscript. We have carefully considered the suggestion of the Reviewer and made some changes. We have tried our best to improve and made some changes to the manuscript.

The red part has been revised according to your comments. Revision notes, point-to-point, are given as follows:

1. The secondary impedance control task will not disturb the primary impedance control task. Authors should provide reason(s) along with relevant literature work.

Thank you for the comment suggested.

We modified throughout the text according to the comment (Above Line 95, page 4). The relevant literature of the text has been listed(Line316-319, page 14).

2. Equation (1) is very fundamental concept and should be added with a citation.

Thank you for the comment suggested.

We modified throughout the relevant literature (Line87, page 3).

3. Authors are highly recommended, they add the table of comparison where they can compare their proposed algorithm with the recently reported works.

Thank you for the comment suggested.

On pages 1 and 2, we give a comparison of the recently reported works, then propose our reverse priority force control. 

4. There are many language/grammatical errors, which should be corrected.

Thank you for the comment suggested.

We modified some language/grammatical errors throughout the manuscript.

5. In the whole manuscript all the figures are just white screens, they are not visible. It may be at the end of the journal portal or submission itself, so I am unable to see any results provided.

Thank you for the comment suggested.

We modified the figures throughout the manuscript.

6. Recent most references are not provided, authors are encouraged to add recent citations to support the study.

Thank you for the comment suggested.

We modified and added the recent references on the lists to support our study.

Kind regards,

Haiyan Liu

Round 2

Reviewer 2 Report

should be accepted.